# Development and Experimental Research of VFTO Measuring Sensor

**DOI:** 10.3390/s23010264

**Published:** 2022-12-27

**Authors:** Zihan Teng, Jun Zhao, Qi Wang, Haonan Lu, Jiangong Zhang

**Affiliations:** State Key Laboratory of Power Grid Environmental Protection, High Voltage Department of China Electric Power Research, Wuhan 430072, China

**Keywords:** VFTO measurement, impedance converter, disconnecting operation, GIS test

## Abstract

Very fast transient overvoltage (VFTO) generated by an operating disconnector is one of the main reasons for electromagnetic disturbance in gas-insulated switchgear (GIS) substations. Generally, the amplitude of VFTO can be used as one of the references for the insulation design of GIS primary electric power equipment, so it is necessary to obtain its accurate amplitude. In this study, a new VFTO measuring sensor is developed and its measurement performance is demonstrated through hundreds of operations by a disconnector in a 220 kV GIS test circuit. The validation shows that the low cut-off frequency of the new VFTO measuring sensor has been greatly expanded to 0.01 mHz, which is improved by about 50% compared with the old sensor. The measurement accuracy of amplitude of VFTO micro-pulse improves greatly by about 80% compared with the old one. Thus, the new VFTO measuring sensor can fully meet the measurement needs of trapped charge voltage, power frequency voltage, and high-frequency transient voltage in VFTO waveform. It can be used to provide more accurate data support for insulation design of GIS primary power electric equipment in extra-high voltage (EHV) and ultra-high voltage (UHV) GIS substations.

## 1. Introduction

Very fast transient overvoltage (VFTO) is one of the main factors affecting the stable operation and insulated intension designs of primary electric power equipment in substations, and it is also the main electromagnetic disturbance source of secondary equipment [1,2,3]. When the disconnector in gas-insulated switchgear (GIS) is operated, a repeated breakdown process occurs between the contact gaps, resulting in a steep pulse traveling wave. The wave propagates along the GIS pipeline and refracts or reflects many times at the place where the wave impedance is discontinuous, and forms very fast transient overvoltage [4,5,6,7]. Especially in extra-high voltage (EHV) and ultra-high voltage (UHV) substations, the high amplitude of VFTO will cause more series faults such as insulation damage to the GIS main circuit and adjacent equipment due to the decrease in insulation margin of the primary electric power equipment [8,9].

It is vital to record the key parameters of VFTO, such as trapped charge voltage, the peak value of VFTO, and frequency spectrum of a transient waveform, during the whole process of disconnecting operations [10,11]. At present, many methods and devices for measuring VFTO have been proposed and developed. Among them, using measuring equipment using differential and integral circuits is difficult to measure VFTO due to the influence of the outer metal shielding structure of GIS basin-type insulator [12,13,14]. In addition, the method of using GIS bushing to construct a VFTO measuring sensor is largely limited by the bushing structure and cannot achieve a high accuracy of the sensor, which makes it difficult to guide the insulation design of GIS [15]. Moreover, using a floating ring electrode embedded in a spacer, as well as the electric field sensor, is also difficult to apply due to the influence of measurement bandwidth and stability [16].

In general, using the capacitive divider sensor is the most effective way to obtain a fast transient voltage signal among all measurement methods. The performance parameters of measurement bandwidth are excellent, with a wide-band cut-off frequency ranging from several hundred Hz to 1 GHz [17,18,19,20]. This kind of sensor is widely used in medium-voltage (MV) networks, which provide accurate voltage measurements for intelligent electronic devices [21,22,23]. At present, it is also widely used in VFTO measurement. Xie et al. used a wide-band parallel resistive–capacitive voltage divider for online monitoring of transient voltages in a 220 kV power grid. The step response 10~90% rise time of this sensor is approximately 29 ns, and the 3 dB bandwidth only covers the range of DC to 10 MHz, which is unable to meet VFTO measurement requirements [17]. Wang et al. used a capacitive voltage divider and differentiating–integrating circuit to make a miniaturized VFTO measurement system, which can be applied to actual VFTO measurements [24]. Moreover, Winkelholz J. et al. presented an improved resistive voltage divider with additional compensation capacities to extend the linear bandwidth, which improved from 115 kHz to 88 MHz [25]. However, it can be found that with the low-frequency cut-off frequency of the resistance–capacitance voltage divider sensor it is often difficult to reach the power frequency 50 Hz, so it is also difficult to meet the measurement requirements of the quasi-DC component in VFTO [14,26].

Furthermore, a method of combining the capacitive divider with an impedance converter was proposed to expand the low-frequency bandwidth. So, the measuring device consisting of a hand-hole type capacitive voltage divider and an impedance converter is called the VFTO measuring sensor. For example, Chen et al. developed an impedance converter and increased the input impedance to 10 MΩ [27], with which the low cut-off frequency of the VFTO measuring sensor reduced to about 50 Hz. Hence, the power frequency components in VFTO have been measured. Another impedance converter was also developed by the previous work of our research group, in which bandwidth ranges from 0.003 Hz to 100 MHz [28]. Therefore, the low frequency bandwidth of the measuring sensor is further expanded. However, it could be found that the trapped charge voltage of VFTO is still not measured accurately during some tests. Moreover, the measurement signal is vulnerable to interference caused by transient ground potential difference. Therefore, it is critical to improve the broadband measurement capability and the anti-interference ability of the VFTO measuring sensor.

In this study, we developed a capacitive voltage divider and new signal conditioning sensor. By comparing the difference between old and new signal conditioning sensors in broadband response and transient interference measurement, it is verified that the performance of the new signal conditioning sensor is better than the old one. It fully meets the measurement needs of the high-frequency transient voltage, power frequency voltage, and trapped charge voltage in the whole process of VFTO waveform. On the other hand, it has a high common mode disturbance rejection ratio and can suppress common mode disturbance signals generated by transient ground potential difference. Thus, VFTO measurement is recommended in the extra-high voltage (EHV) and ultra-high voltage (UHV) GIS substations.

This paper is organized as follows. In Section 2, the development of the capacitive voltage divider is introduced. Then, the parameter design and signal integrity analysis of the signal conditioning circuit are also described. In Section 3, the synchronous measurement results in the time domain of old and new signal conditioning sensors on a 220 kV GIS test platform are compared, and the full bandwidth spectrum analysis results from quasi-DC to 300 MHz are also discussed. Finally, the conclusions are drawn in Section 4.

## 2. Design of VFTO Measuring Sensor

### 2.1. Capacitive Voltage Divider

The capacitive voltage divider is the first part of the VFTO measuring sensor, and its structure is shown in Figure 1. It uses the stray capacitance between the hand-hole electrode and the high-voltage busbar to form high-voltage port capacitance and uses the stray capacitance between the electrode and the grounded hand-hole cover plate to form the low-voltage port capacitance. The handhole is equivalent to introducing a singular point into the uniform double conductor transmission lines structure of GIS where the wave impedance changes significantly. On the one hand, the smaller the diameter and depth of the handhole, the more conducive it is to reducing the interference of high-order mode electromagnetic waves on VFTO measurement. On the other hand, the larger the size of the hand-hole electrode, the more likely it is to generate significant fluctuations and oscillations on the electrode under the influence of parasitic parameters. Moreover, if the size of the handhole and electrode is too small, it is not conducive to engineering design. Therefore, it is necessary to reasonably design the structure of the capacitive voltage divider.

Generally, the size of electrode, as well as the shape of handhole, will all affect the capacitance parameters of the high-voltage port and low-voltage port of the capacitive voltage divider. It can also affect the wave front rise process of the measured waveform. Therefore, it is critical to determine the electrode size. Usually, the wave front rise time of VFTO transient pulse is about 3 ns, during which the transient wave can travel 900 mm. To ensure that the change of wave-head amplitude at the distance of electrode diameter meets the 5% accuracy requirement of engineering measurement, the electrode diameter is required to be less than 45 mm. The diameter of the handhole shall be as small as possible after meeting the condition that the electrode can be installed in it. Furthermore, in order to meet the requirements of voltage amplitude with several volts at the measuring port of secondary equipment, it is necessary to determine the voltage division ratio of the sensor. According to GIS circuits with different voltage levels, the amplitude of VFTO varies from hundreds of kilovolts to thousands of kilovolts. Therefore, the voltage division ratio of the capacitor voltage divider is usually set within the range of 1/1,000,000 to 1/100,000. Since the capacitance parameters are affected by the size of the handhole and diameter of the electrode, it is also necessary to adjust and optimize these parameters after determining the voltage division ratio, so as to achieve the purpose of mutual coordination between them. Using the COMSOL Multiphysics simulation software, these parameters are obtained through electrostatic field simulation. According to the GIS voltage level used in our study, the voltage division ratio of the capacitor voltage divider is set at around 1/200,000. Hence, the stray capacitance parameters between the high-voltage busbar, the hand-hole electrode and the hand-hole cover plate, as well as the size of the electrode and handhole, are simulated and shown in Table 1.

### 2.2. Signal Conditioning Sensor

The signal conditioning sensor is the second part of the VFTO measuring sensor, and it is used to optimize the output signal of the secondary port of the capacitive voltage divider. It can realize impedance transformation, common mode disturbance suppression, and signal amplitude scaling, so as to improve the low-frequency transmission performance of the capacitive voltage divider and optimize the output signal. Theoretically, the output signal of the secondary port of the capacitive voltage divider and the input signal at the high-voltage port meet the following transmission characteristics:(1)$$\frac{V_{out}}{V_{in}}=\frac{C_{1}}{C_{1}+C_{2}}$$
where $C_{1}$ is the high-voltage port capacitance and $C_{2}$ is the low-voltage port capacitance. Since the transfer function is independent of frequency, the capacitive voltage divider can be used to measure voltage signals of any frequency. However, the sampling equipment usually connected to the low-voltage port of the capacitive divider is equivalent to a resistive load, so the signal acquisition system based on the capacitive divider can be equivalent to the circuit shown in Figure 2. Therefore, the relationship between the output signal and the input signal of the VFTO measuring sensor satisfies the following formula:(2)$$\frac{{\overset{•}{V}}_{out}}{{\overset{•}{V}}_{in}}=\frac{j\omega C_{1}R_{L}}{1+j\omega \left(C_{1}+C_{2}\right)R_{L}}\approx \frac{j\omega C_{1}R_{L}}{1+j\omega C_{2}R_{L}}=\frac{C_{1}}{\frac{1}{j\omega R_{L}}+C_{2}}$$
(3)$$\left|\frac{V_{out}}{V_{in}}\right|=\frac{\omega C_{1}R_{L}}{\sqrt{1+{\left(\omega C_{2}R_{L}\right)}^{2}}}=\frac{C_{1}}{\sqrt{{\left(\frac{1}{\omega R_{L}}\right)}^{2}+{\left(C_{2}\right)}^{2}}}$$
where $R_{L}$ is the equivalent resistive load and ω is the angular frequency.

The relationship between input and output voltage shown in Formula (3) is described in Figure 3, while $C_{1}$ and $C_{2}$ are the results shown in Table 1, respectively. Regarding the $1/\omega R_{L}$ as a variable, we observe that the amplitude of the sampled signal will change with the frequency when the low-voltage port capacitance $C_{2}$ will not meet the condition that it is far greater than $1/\omega R_{L}$.

In order to ensure that the sampling signal size is not affected by the frequency change, the low-voltage port capacitance $C_{2}$, the equivalent load $R_{L}$, and the signal frequency should meet the following formula:(4)$$f\gg \frac{1}{2\pi R_{L}C_{2}}$$

The frequency f in Formula (4) usually represents the 3 dB low-frequency cut-off frequency of the VFTO measurement sensor. Therefore, the equivalent load resistance $R_{L}$ can be reasonably designed according to the requirements of the measurement frequency band and the low-voltage port capacitance $C_{2}$. To measure the quasi-DC component of VFTO, the resistance $R_{L}$ should be large enough to reach the order of GΩ.

In this paper, a large input impedance $R_{L}$ is obtained by using the virtual break characteristic of an operational amplifier. The input impedance of the new signal conditioning sensor can reach 250 GΩ measured by an impedance network analyzer, which is far greater than the 40 GΩ input impedance of the old impedance converter previously developed by our research group, as mentioned in Ref. [28]. Therefore, its low-frequency bandwidth can reach 50 μHz, which is significantly improved. The schematic diagrams of the new signal conditioning circuit and the old one are shown in Figure 4. Considering the harsh electromagnetic disturbance measurement environment of the signal conditioning sensor, a differential impedance conversion signal conditioning circuit was developed according to the design idea of the instrument amplifier circuit. The layout and parameter selection of this differential circuit are all completed in Multisim simulation software. The amplitude frequency and phase frequency transfer characteristics of the new signal conditioning circuit are shown in Figure 5. It shows that the amplitude frequency characteristics of the designed circuit only vary between ±0.1 dB in the range of 0.01 mHz~160 MHz. Its 3 dB cut-off frequency can reach 190 MHz, and it has a linear change in phase frequency characteristics within 0.01 mHz~250 MHz, totally meeting the wide-band measurement requirements of VFTO. Moreover, the common mode disturbance rejection ratio of the new signal conditioning sensor can reach 53 dB, which is much higher than that of the old impedance transformation circuit, effectively restraining the coupling of common mode disturbance of ground potential difference.

Figure 6 shows the transient electric field on the whole circuit board surface and the transient electric field induced on the internal surface of the metal case generated by the transient voltage from the input port of the signal conditioning sensor. In order to visually represent the size of the radiation field, we perform logarithmic operation on the amplitude of the electric field to convert the unit of the field strength into the decibel value. We notice that the transient signal induces a strong transient field around the circuit, which almost covers the entire circuit board. The maximum amplitude of the transient field around the circuit can reach 82.2 dB, which will decay to about −20 dB only in the four corners of the circuit board. Similarly, the electromagnetic field radiates from the patch pin on the surface of the circuit board to the internal space of the device, and the intensity of the induced field on the surface of the metal shell is usually about 20 dB. Figure 6b shows that the amplitude of the surface electric field generated by the transient signal in each part of the circuit is basically the same, and a large surface electric field can only be excited when passing through some small apertures. The maximum value of the surface electric field is about 82.2 dB, and the minimum value is about 122 dB, which is located on the lower surface of the circuit board. Therefore, in order to reduce the interference of the surface electric field, the use of vias with too small an aperture will be avoided when connecting circuits in the circuit board design. Moreover, long and thin leads will also be avoided, so as to reduce the large surface radiation electric field.

Furthermore, the response change characteristics of the signal conditioning circuit in the range of 1 MHz to 100 MHz are calculated, as shown in Figure 7. It can be found that the amplitude of the signal in the frequency band range of 1 MHz to 10 MHz has only 0.075 dB attenuation. The −3 dB high-frequency cut-off frequency of the circuit is about 95 MHz, which indicates that this signal conditioning sensor can meet the measurement requirements of different components in VFTO signal.

The complete manufactured prototype of the new signal conditioning sensor is shown in Figure 8. Moreover, a lithium battery charging and discharging circuit module and DC voltage stabilizing filter module have been integrated in the signal conditioning sensor, which can provide a stable and low-noise ±5 V DC power supply for the operational amplifier. Through an experimental test, the signal conditioning sensor can be used continuously for 13 h when the battery is fully charged.

As shown in Figure 9, we obtained the amplitude frequency and phase frequency transfer characteristics of the new signal conditioning sensor through a sweep-frequency test in the laboratory. We notice that the 3 dB low cut-off frequency of this sensor can reach 0.01 mHz at least and its 3 dB high cut-off frequency can also reach 90 MHz. The amplitude–frequency response is stable which means it changes not more than ±0.1 dB in the range of 0.1 mHz to 20 MHz. In general, this experimental result is in good agreement with the simulation results shown in Figure 5 and Figure 7.

## 3. VFTO Measurement Results

In this study, about 500 VFTO measurement experiments were carried out during disconnecting operations on a 220 kV GIS circuit, and the actual characteristics of the VFTO measuring sensor were verified. The 220 kV GIS circuit is composed of two disconnectors (DS1 and DS2), two bushings (AC side bushing and DC side bushing), and a long and straight GIS pipeline, as shown in Figure 10. Among them, the DS1 is the main disconnector used to generate VFTO by repeated operations, and the DS2 is an auxiliary disconnector, which is always disconnected. The AC side bushing is used to connect with an AC power supply with amplitude of 179 kV and frequency of 50 Hz. The DC side bushing is usually in a no-load state. During the tests, the VFTO measurement system is installed at M1 point in the GIS pipeline. This measurement point is located at the end of the no-load short bus between DS1 and DS2. Therefore, the measured results can simultaneously reflect the quasi-DC component, power frequency component, and high-frequency transient component in VFTO.

The overall structure of the VFTO measurement system is shown in Figure 11, which mainly includes five parts: a capacitive voltage divider, signal conditioning sensor, electrical–optical signal converter, field acquisition oscilloscope, and remote-control computer. The input port of the signal conditioning sensor is directly connected to the secondary output port of the capacitive voltage divider, and its output port is directly connected to the field acquisition oscilloscope. The oscilloscope was powered by a lithium battery through a DC–AC inverter to avoid ground potential difference conduction disturbance, and it communicated with a computer through a fiber-optic communication system to realize remote control. These devices were all installed in the shielding box to avoid the influence of transient disturbance during disconnector operations.

We use the impedance converter sensor developed in the early work of our research group mentioned in Ref. [28] to synchronously measure VFTO signals with the new type, so as to compare the difference in measurement results between them. The typical whole waveform of VFTO measured by these two sensors are shown in Figure 12 and Figure 13. We observe that the VFTO obtained by the new measuring sensor have roughly the same characteristics as the typical VFTO waveforms given in the Refs. [4,9], which means that the measuring sensor can completely record VFTO signals. 

As shown in Figure 12, the VFTO waveform on the load side of GIS is composed of a 50 Hz power frequency sine wave, dozens of step waves and high-frequency oscillation waves superimposed at the beginning of each step wave. The time interval between these high-frequency oscillation pulses shows the characteristics of being first dense and then sparse during the opening operation of disconnector. In the closing operation, it shows the opposite characteristics. Combined with the description in Section 2.2, the signal conditioning sensor is used to improve the low-frequency measurement performance of the capacitive voltage divider and optimize the high-frequency transient signal. Hence, the characteristics of the step waves and the fast high-frequency oscillation pulses in the VFTO measurement results are compared in our research, respectively.

The step waves represent the quasi-DC components in the VFTO signal, which mainly reflects the variation characteristics of trapped charge in the no-load short busbar within the time interval between two adjacent high-frequency oscillation pulses. During this time, the voltage on the no-load short busbar consists of two parts: the power frequency voltage induced by the stray capacitance between disconnector gap, and the trapped charge voltage represented by the trapped charge in the no-load short busbar. Usually, the power-frequency-induced voltage is far less than the trapped charge voltage. Therefore, the change in step waves can be approximately equivalent to the change in trapped charge voltage. Since the trapped charge in the no-load short busbar is difficult to release in the time interval between two adjacent high-frequency oscillation pulses, this voltage component is theoretically equal to the power frequency voltage at the end of the breakdown process and remains unchanged. Hence, the low-frequency measurement performance of the signal conditioning sensor can be compared by comparing the change characteristics of the step waves in the VFTO measurement results.

In our study, the performance of the signal conditioning sensor is compared by calculating the proportional coefficient k of trapped charge voltage attenuation. As shown in Figure 14, the attenuation proportional coefficient k of trapped charge voltage is defined as the change degree of trapped charge voltage at the beginning of the next high-frequency oscillation pulse in the two adjacent high-frequency oscillation pulses relative to the trapped charge voltage at the end of the last high-frequency oscillation pulse. Its formula is as follows:(5)$$k=\left|\frac{U_{end}-U_{start}}{U_{start}}\right|\times 100\%$$

Nearly 500 groups of attenuation proportional coefficients were counted, and the results are shown in Figure 15. The black triangle symbol represents the results of the old impedance converter defined as $k_{1}$, and the red circle symbol represents the results of the new signal conditioning sensor defined as $k_{2}$. We notice that the statistical eigenvalues of $k_{1}$ approximately distribute between 5% and 15%, and there is also a very bad situation in which some eigenvalues of $k_{1}$ reach 45%. However, the statistical eigenvalues of $k_{2}$ distribute between 0.2% and 2.5%, which are more concentrated and smaller than that of $k_{1}$. Generally, the distribution of $k_{1}$ is relatively scattered without strong regularity, but the distribution of $k_{2}$ is very dense and more uniform with a smaller dynamic error range. It can be easily concluded that the smaller the eigenvalues, the less the attenuation of trapped charge voltage. Therefore, the new signal conditioning sensor has a lower and more stable low-frequency characteristic than the old impedance converter.

The fast high-frequency oscillation pulse reflects the high-frequency transient performance of the signal conditioning sensor. Expanding the full waveforms, the typical waveforms of the fast high-frequency oscillation pulse can be observed in Figure 16. Among them, the red curve represents the measurement result of the new signal conditioning sensor, and the black curve represents the measurement result of the old impedance converter. From the time-domain process of these oscillation pulses, it can be found that the overall change trend of these waveforms is almost consistent. After a rapid rise within several nanoseconds, the high-frequency oscillation pulse turns into a damping oscillation attenuation process with a fixed oscillation frequency of about 10 MHz and attenuates to zero after several microseconds. However, it can be found that the result of the rapid oscillation process is much different within 0.3 microseconds in the beginning of the transient pulse. In the measurement results of the impedance converter, the wave peak amplitude at 0.1 μs moment and the wave valley amplitude at 0.15 μs moment have dramatic changes, which is caused by the superposition of multiple small oscillation waves with a frequency of more than 70 MHz on the main oscillation waveform with a frequency of 10 MHz. These tiny oscillation waveforms almost lasted several microseconds before gradually decaying to zero. However, in the measurement results of the signal conditioning sensor, it is difficult to find such small oscillation waveforms with a frequency of more than 70 MHz. 

Furthermore, the fast Fourier transform (FFT) is used to obtain the spectrum characteristics of the high-frequency oscillation pulse to distinguish the difference in measurement results. As shown in Figure 17, the measurement results of these two sensors are almost the same at 1.2 MHz, 4.9 MHz, and 10.2 MHz, respectively. However, the results are obviously different at the frequencies of 73 MHz, 87 MHz, 254 MHz, and 275 MHz.

In order to illustrate the accuracy of measurement results, we used EMTP simulation software to model this 220 kV GIS circuit and simulated the micro-pulse waveform of VFTO. The time domain and frequency domain simulation results are shown in Figure 18. From Figure 18, we observe that the simulation results are similar to the measurement results of the new sensor both in the time domain and frequency domain. The transient pulse reached its peak value at 0.5 μs. Compared with the measurement results of the old type, there are also no rapid high-frequency oscillations with a frequency higher than 70 MHz within 1 μs. Therefore, we believe the measurement results of the new VFTO measuring sensor are accurate and reliable.

By designing an IIR low-pass filter in MATLAB and filtering the time domain waveform with 70 MHz as the boundary, two transient waveforms with frequencies less than 70 MHz and higher than 70 MHz are obtained, respectively, as shown in Figure 19. From Figure 19b, we notice that the amplitude of the oscillation component with frequencies greater than 70 MHz can reach 100 kV in the measurement results of the old impedance converter. This is almost four times the measurement result of the new sensor. Figure 20 shows the spectrum characteristics of these two time-domain waveforms. It can be found that the two transient waveforms with filtered frequency less than 70 MHz almost coincide, indicating that the two types of measurement sensors have almost the same measurement performance in the 70 MHz frequency band. 

Combined with the equivalent circuit of the VFTO measuring sensor shown in Figure 2, the capacitance at the low-voltage port of the capacitive voltage divider, the inductance of the measuring cable, and the capacitance at the input port of the sensor can easily form a resonant circuit, and its resonant frequency can be expressed as:(6)$$f=\frac{1}{2\pi \sqrt{\frac{\left(C_{1}+C_{2}\right)C_{L}L_{W}}{C_{1}+C_{2}+C_{L}}}}$$
where $C_{L}$ is the capacitance at the input port of the sensor, and $L_{W}$ is the equivalent inductance of the measuring cable. When the capacitance at the input port of the sensor corresponds to the common mode capacitance with 1.3 pF and the differential mode capacitance with 0.1 pF, the resonant frequencies of this circuit are 73 MHz and 256 MHz. These two frequency components exactly correspond to the high-frequency components in the measurement results of the old impedance converter, as shown in Figure 20. Since the new signal conditioning sensor has a strong common mode disturbance suppression capability, it can suppress the common mode disturbance signal generated by circuit resonance. Therefore, it can be sure that the measurement results of the new signal conditioning sensor are more accurate.

## 4. Conclusions

A full set of the VFTO measuring sensor is developed in this paper, which includes a capacitive voltage divider, a signal conditioning device with impedance transformation and signal optimization capabilities, a data acquisition card for signal sampling, and a computer for remote control. In this research, the structure of the capacitive voltage divider and the circuit of impedance transformation signal conditioning sensor are designed through simulation. Moreover, the measurement performance of this sensor is verified by hundreds of disconnecting operations in a 220 kV GIS circuit, and the conclusions are as follows:
(1)The method, using the virtual break characteristic of operational amplification to obtain very large input impedance, is very effective in expanding the low-frequency measurement bandwidth of the impedance converter. Thus, the time constant of the equivalent circuit of the low-voltage port of the capacitor divider is greatly extended, and an extremely low cut-off frequency is obtained. In this research, the input impedance of the impedance transformation signal conditioning device was increased from 40 GΩ to 250 GΩ, thereby reducing the low-frequency cut-off frequency from the original 10 mHz to 0.05 mHz. The VFTO measurement sensor can fully meet the measurement requirements of residual voltage components. Moreover, a differential circuit with a common mode disturbance rejection ratio of 53 dB is designed to effectively suppress the common mode disturbance signal caused by the transient ground potential difference in the measurement signal.(2)The transient surface electric field of the signal conditioning circuit board is simulated by COMSOL Multiphysics software. The simulation results show that the transient signal excites a large surface electric field around the wires of the circuit board. The surface electric field gradually attenuates from the center of the circuit board to the edge, which can reach 82.2 dB at the center. In the design of this signal conditioning circuit board, the smaller the wire width is, the greater the surface electric field will be excited. To avoid crosstalk between wires, the width of it should be at least greater than 10 mil. In addition, vias with too small an aperture will significantly stimulate the nearby surface electric field, so it is recommended that the aperture of vias should be at least greater than 5 mil. In addition, the amplitude frequency transfer characteristics between the input and output ports of the circuit board are also calibrated through a sweep-frequency experiment. The results show that the high cut-off frequency of the signal conditioning circuit board is about 95 MHz, which is close to the simulation results. The fluctuation in amplitude frequency transfer characteristics is less than 0.1 dB in the range of 1 MHz to 10 MHz. Therefore, it can be concluded that the signal conditioning circuit can meet the measurement requirements of fast transient components of VFTO signals.(3)Hundreds of disconnecting operation experiments are carried out on a 220 kV GIS test platform to verify the actual performance of the VFTO measuring sensor. The results show that the low-frequency performance of the new signal conditioning sensor is far better than that of the old impedance converter, and it can measure the trapped charge voltage component in VFTO well. The relative error of the measurement results is less than 2.5%, and the measurement accuracy is improved by at least 50% compared with the old sensor. The VFTO results measured by the old impedance converter are superimposed with high-frequency interference signals generated by internal circuit resonance. Although the signal is only about 500 mV at the output port of the sensor, it will cause a deviation of up to 100 kV in the measurement results after being converted into the transient overvoltage in the bus according to the voltage division ratio. After filtering to eliminate high-frequency resonance interference, we notice that the measurement results of the new and old sensors are almost identical. This shows that the new signal conditioning sensor can use the advantage of the differential circuit to effectively suppress the disturbance signal generated by internal circuit resonance, so as to obtain a more accurate measurement of VFTO amplitude. Therefore, the VFTO measurement result of the new sensor can be used to guide the insulation design of GIS primary equipment more accurately.

In general, engineers can use this VFTO measuring sensor to accurately measure the VFTO signal generated during the disconnecting operations in GIS substations, which is very helpful in the insulation design of primary electric power equipment in GIS substations. Moreover, after obtaining accurate VFTO amplitude and frequency characteristics through the VFTO measurement sensor, targeted transient interference suppression methods can be used to achieve more accurate anti-interference protection for secondary equipment, which is conducive to reducing energy waste and achieving the construction goal of clean and environment-friendly power grids.

## Figures and Tables

**Figure 1 sensors-23-00264-f001:**
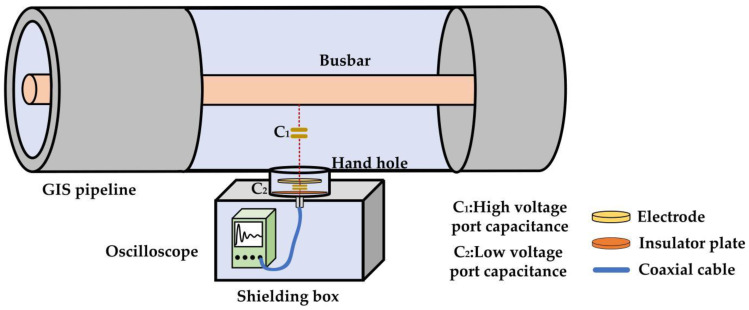
The structure diagram of hand-hole capacitive voltage divider sensor.

**Figure 2 sensors-23-00264-f002:**
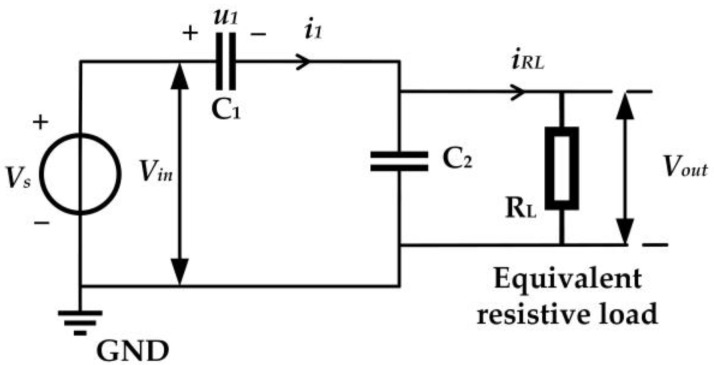
Schematic diagram of VFTO measurement sensor.

**Figure 3 sensors-23-00264-f003:**
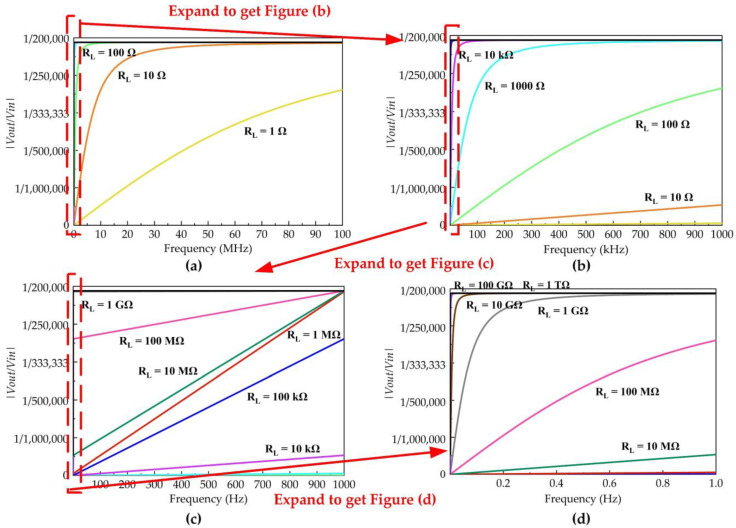
The Influence of frequency *f* and equivalent resistive load $R_{L}$ on the ratio of input voltage to output voltage. (**a**) Equivalent resistive load $R_{L}$ changes from 1 Ω to 10 Ω; (**b**) Equivalent resistive load $R_{L}$ changes from 10 Ω to 10 kΩ; (**c**) Equivalent resistive load $R_{L}$ changes from 10 kΩ to 1 GΩ; (**d**) Equivalent resistive load $R_{L}$ changes from 10 MΩ to 10 TΩ.

**Figure 4 sensors-23-00264-f004:**
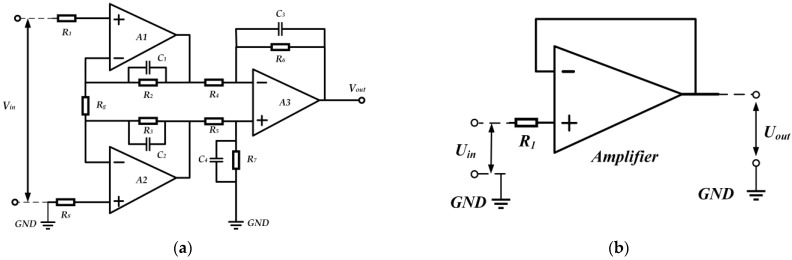
Schematic diagram of signal conditioning circuit. (**a**) The new circuit; (**b**) the old circuit.

**Figure 5 sensors-23-00264-f005:**
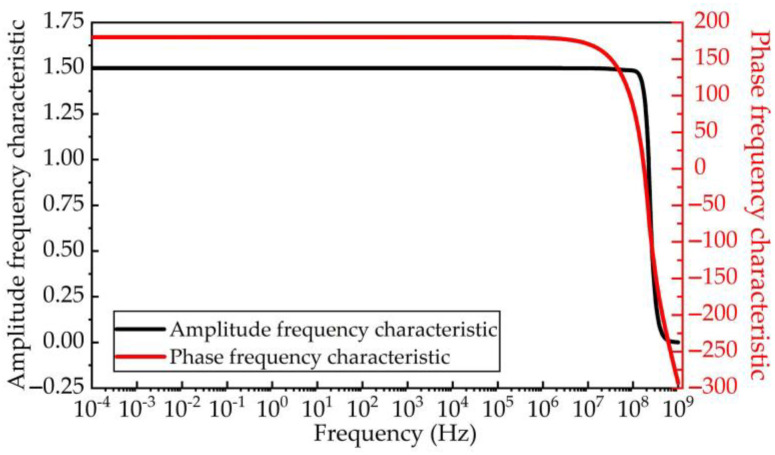
The amplitude frequency characteristic and phase frequency characteristic of signal conditioning circuit.

**Figure 6 sensors-23-00264-f006:**
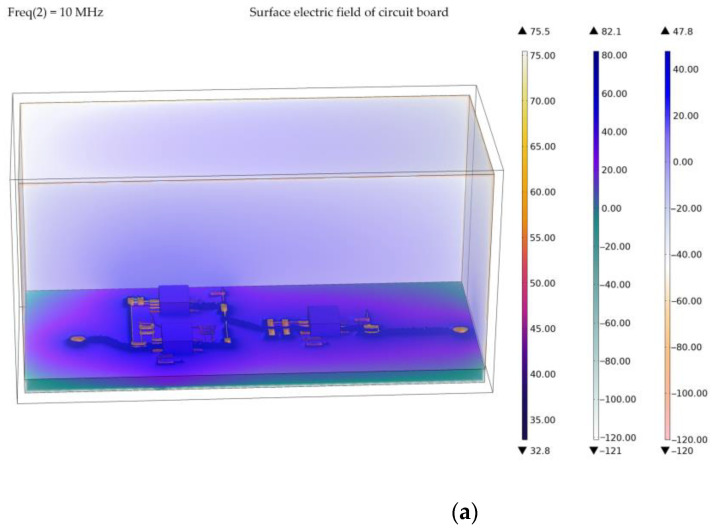
Simulation results of surface electric field of signal conditioning circuit board under transient signal input. (**a**) Surface electric field on the inner surface of signal conditioning sensor; (**b**) Electric field on the surface of signal conditioning circuit board.

**Figure 7 sensors-23-00264-f007:**
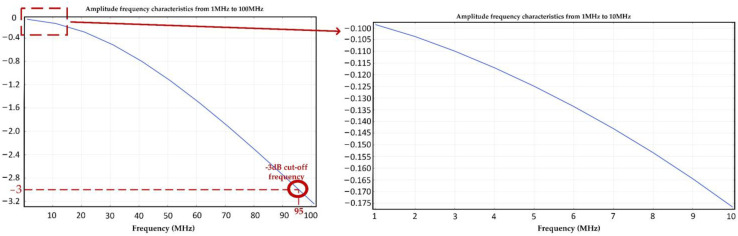
The response characteristics of signal conditioning circuit in the range of 1 MHz to 100 MHz simulated by COMSOL Multiphysics software.

**Figure 8 sensors-23-00264-f008:**
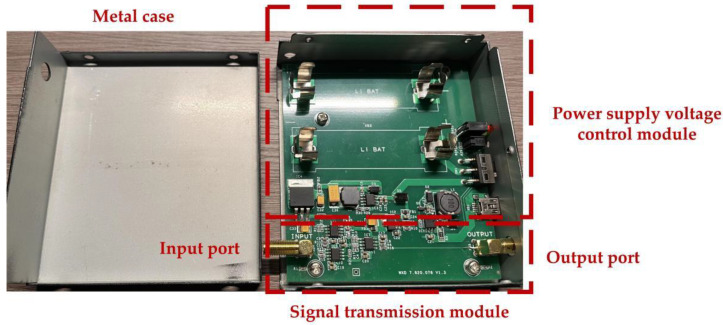
Complete manufactured prototype of new signal conditioning sensor.

**Figure 9 sensors-23-00264-f009:**
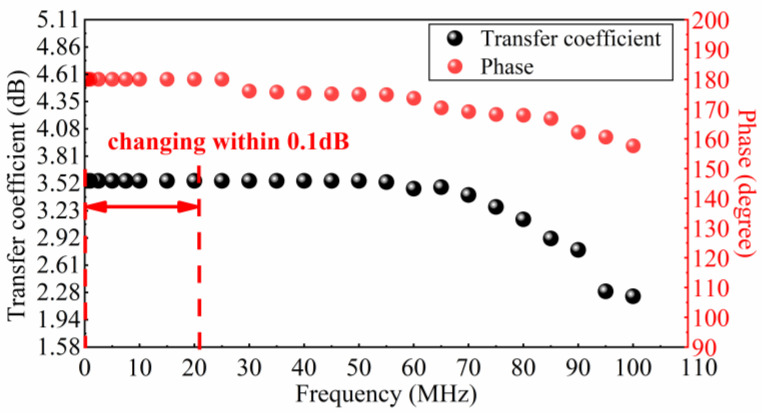
The sweep-frequency calibration result of the new signal conditioning sensor.

**Figure 10 sensors-23-00264-f010:**
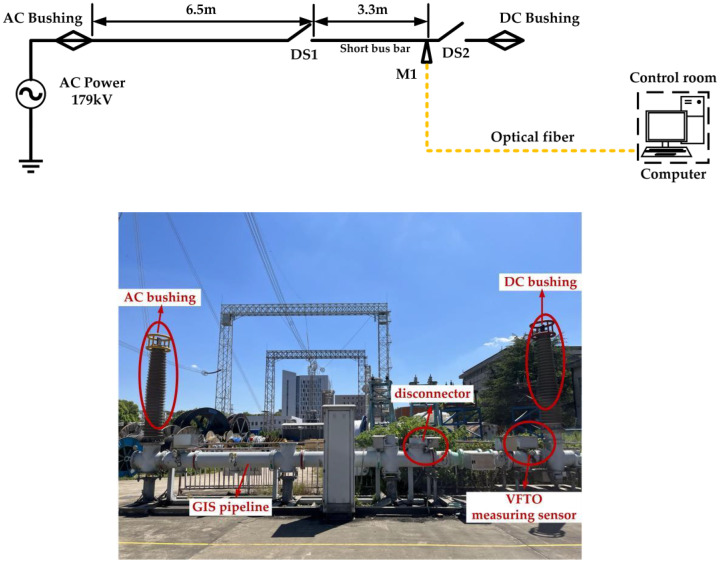
Schematic diagram of the 220 kV GIS test circuit.

**Figure 11 sensors-23-00264-f011:**
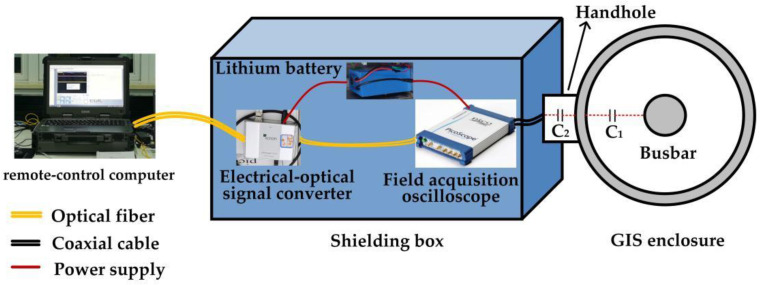
The overall structure of the VFTO measurement system.

**Figure 12 sensors-23-00264-f012:**
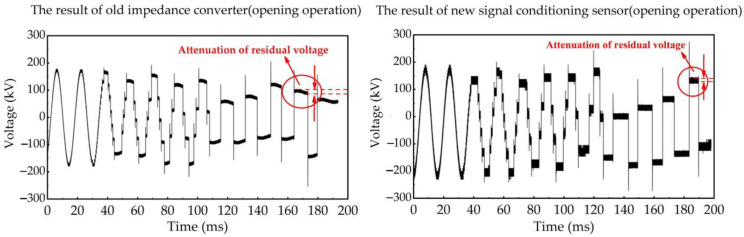
The typical full-time waveform of VFTO measured by the new signal conditioning sensor and the old impedance converter during opening operation.

**Figure 13 sensors-23-00264-f013:**
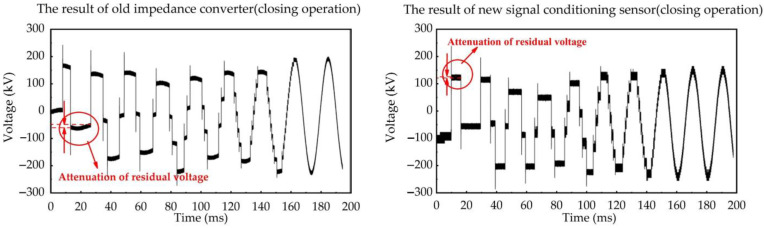
The typical full-time waveform of VFTO measured by the new signal conditioning sensor and the old impedance converter during closing operation.

**Figure 14 sensors-23-00264-f014:**
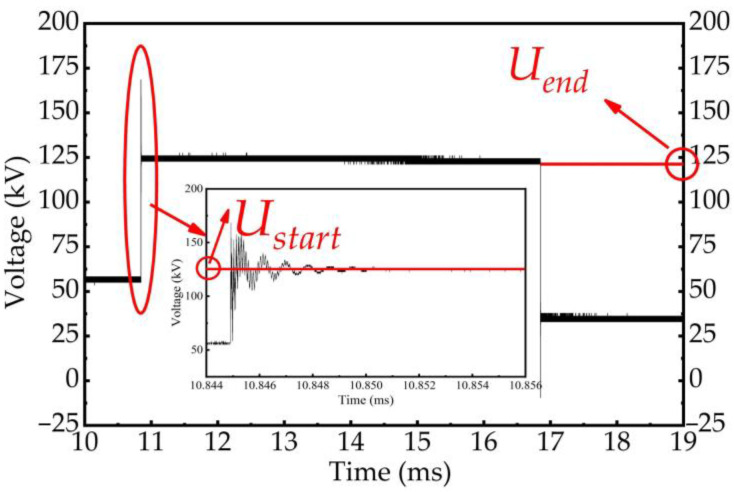
The attenuation changes of trapped charge voltage within the occurrence time of two adjacent high-frequency oscillations.

**Figure 15 sensors-23-00264-f015:**
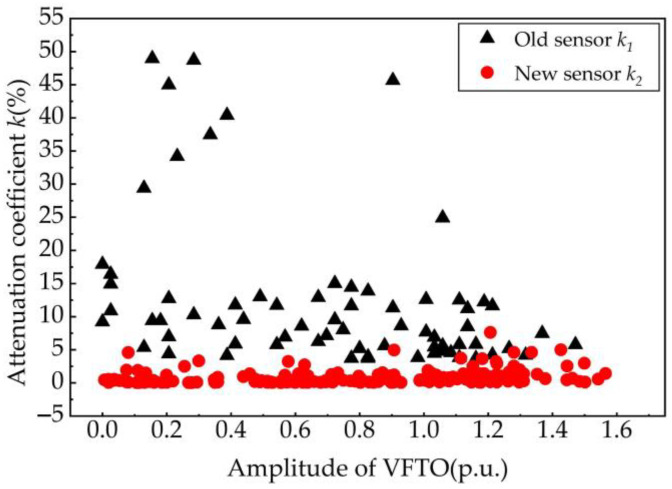
Attenuation proportional coefficient of trapped charge voltage component in measurement results of the new signal conditioning sensor and the old impedance converter.

**Figure 16 sensors-23-00264-f016:**
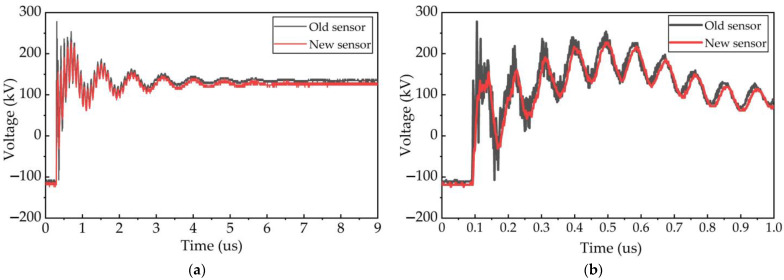
The typical micro pulse waveform of VFTO. (**a**) The whole process waveform of micro pulse; (**b**) the micro pulse waveform within one microsecond.

**Figure 17 sensors-23-00264-f017:**
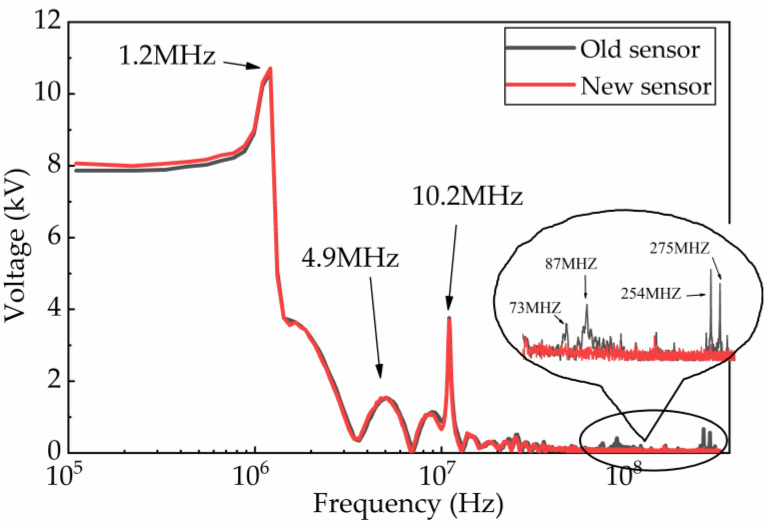
Full bandwidth spectrum of VFTO.

**Figure 18 sensors-23-00264-f018:**
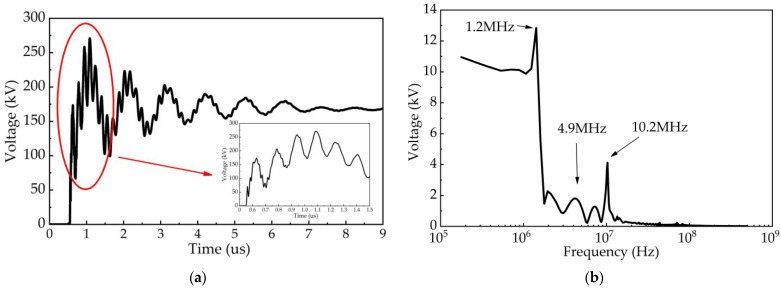
The simulation results of VFTO in time domain and frequency domain. (**a**) Time-domain simulation results; (**b**) Frequency-domain simulation results.

**Figure 19 sensors-23-00264-f019:**
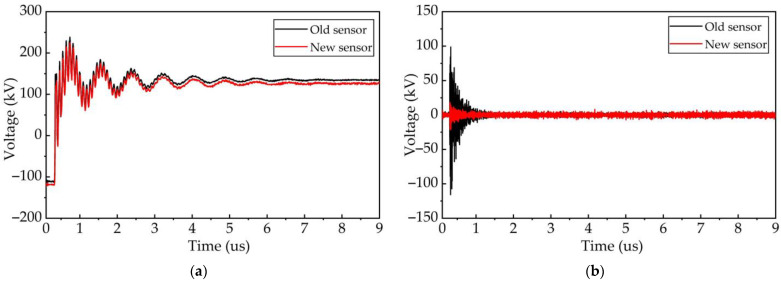
Time domain waveform of VFTO. (**a**) VFTO time domain waveform with frequency lower than 70 MHz; (**b**) VFTO time domain waveform with frequency higher than 70 MHz.

**Figure 20 sensors-23-00264-f020:**
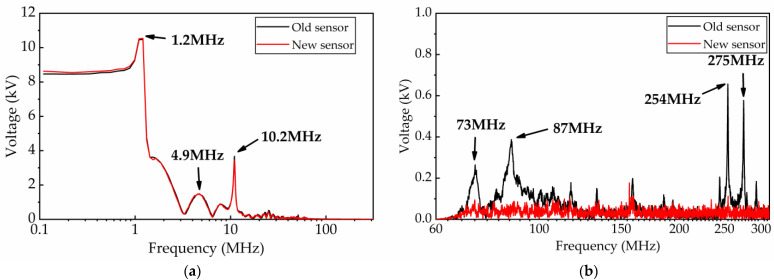
Frequency domain waveform of VFTO. (**a**) Frequency domain waveform with frequency lower than 70 MHz; (**b**) frequency domain waveform with frequency higher than 70 MHz.

**Table 1 sensors-23-00264-t001:** The parameters of port-hole type capacitive divider sensor.

Parameter	Type	Unit
High-voltage port capacitance	0.0537	pF
Low-voltage port capacitance	11.0031	nF
Electrode diameter	40	mm
Hand-hole diameter	50	mm
Hand-hole depth	20	mm

## Data Availability

Not applicable.

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
