# Peer review of "Development and Experimental Research of VFTO Measuring Sensor"

_sensors, 2022, doi:10.3390/s23010264_

Round 1
Reviewer 1 Report
Overall this is a good paper with simulation, circuit design, sensor design and experimental results.
A major issue is the clarification of the circuit in Figure 3.
1. Have the authors considered the bias current and offset voltage of the opamp? How do you make sure the opamp you used have the same parameter (they will have small differences even with the same batch).
2. How the previous circuit differ in performance? I think people can easily think of the idea of large input impedance of an opamp.
Minor issues :
Format issues such as “Author 1, A.B.; Author 2, C.D. Title of the article. Abbreviated Journal Name Year, Volume, page range.”
Still in the reference section. This should be deleted in formal paper.
Author Response
Thank you for your kind comments and valuable time. We provide a point-by-point response to the your comments.
Please see the attachment.

Reviewer 2 Report
The manuscript by Teng et al. presents a new set of very fast transient overvoltage (VFTO) measuring sensors. The structure, design considerations, and simulation methods are well described for each component of the sensor. In addition, the measurement shows significant improvement in the sensor compared to the previous generation by the same team through hundreds of disconnector operation tests. The reviewer believes the results are of importance to the community and interesting to the readers of Sensors. Therefore, it can be considered for publication on Sensors after revising the manuscript with respect to the following comments.
1. In line 78-79, the authors mentioned the previous research results of their research group. The details or citation of the previous research should be included for better readability.
2. Please add the references for COMSOL Multiphysics simulation software
Author Response

(The authors gave the same response as above.)

Reviewer 3 Report
In this work, a new VFTO measuring sensor was developed through Multisim simulation software and COMSOL Multiphysics simulation software, and it was applied to a 220 kV GIS test circuit to verify its measurement performance. The low cut‐off frequency of the new VFTO measuring sensor has been greatly expanded to 0.01mHz, and the amplitude measurement accuracy of VFTO improved about 80%, which can be used to provide more accurate data support for insulation design of GIS primary power electric equipment in extra‐high voltage (EHV)and ultra‐high voltage (UHV) GIS substations.
1. It is suggested to give the detailed differences between the new and old signal conditioning circuit as shown in Fig.3.
2. In this work, the authors claimed that “a new VFTO measuring sensor was developed through Multisim simulation software and COMSOL Multiphysics simulation software”, but not mention the influence of the input paraments of the input impedance and impedance conversion signal conditioning circuit on the low cut‐off frequency and the amplitude measurement accuracy.
3. The relationship between the experimental results and the simulation results should be analysis in the text. If there is deviation for experimental results?
Author Response

(The authors gave the same response as above.)

Round 2
Reviewer 1 Report
I have no further comments